# Assessment of Migration of the Urethral Bulking Agent Zhoabex G^®^ from the Urethral Injection Site to the Distant Organs in a Rabbit Model

**DOI:** 10.3390/ijms262110286

**Published:** 2025-10-22

**Authors:** Bhagath Kumar Potu, Diaa Rizk, Muna Aljishi, Ameera Sultan, Wael Amin Nasr El-Din, Stefano Salvatore, Safa Taha

**Affiliations:** 1Department of Anatomy, College of Medicine and Health Sciences, Arabian Gulf University, Manama P.O. Box 26671, Bahrain; potubk@agu.edu.bh (B.K.P.); waela@agu.edu.bh (W.A.N.E.-D.); 2Department of Obstetrics and Gynaecology, College of Medicine and Health Sciences, Arabian Gulf University, Manama P.O. Box 26671, Bahrain; diaarizk@agu.edu.bh; 3Princess Al Jawhara Center for Molecular Medicine, Genetics and Inherited Diseases, Department of Molecular Medicine, College of Medicine and Health Sciences, Arabian Gulf University, Manama P.O. Box 26671, Bahrain; munajma@agu.edu.bh (M.A.); ameeraa@agu.edu.bh (A.S.); 4Department of Human Anatomy and Embryology, Faculty of Medicine, Suez Canal University, Ismailia 41522, Egypt; 5Obstetrics and Gynaecology Unit, Vita-Salute San Raffaele University, IRCCS San Raffaele Hospital, 20132 Milan, Italy; stefanosalvatore@hotmail.com

**Keywords:** hyaluronic acid, urethral bulking agent, stress urinary incontinence, migration, RT-PCR, ELISA, rabbit model, Zhoabex G^®^

## Abstract

Hyaluronic acid (HA)-based urethral bulking agents are promising for the treatment of stress urinary incontinence (SUI), but migration risks to distant organs remain a concern. This study evaluated the migration and cytotoxicity of Zhoabex G^®^, an HA-based bulking agent, in a female rabbit model. Twenty-seven female New Zealand white rabbits were randomized into control (no injection), sham (saline), and experimental (Zhoabex G^®^) groups (*n* = 9 each). After 5 months, tissues from the kidney, lung, liver, and spleen were analyzed using quantitative RT-PCR for hyaluronan synthase (HAS1, HAS2, HAS3) and hyaluronidase (HYAL2) gene expression, and ELISA for HA concentrations. No significant differences in gene expression were observed across groups (*p* ≥ 0.05, range: 0.166–0.997), with experimental fold change values near sham baselines (e.g., kidney HAS2: 0.987 ± 0.071, *p* = 0.422). Similarly, HA concentrations showed no group differences (*p* = 0.577; e.g., kidney: 119.2–121.8 ng/mL), reflecting organ-specific basal levels. These findings indicate that Zhoabex G^®^ remains localized at the urethral injection site, with no evidence of migration or cytotoxicity in distant organs. The biodegradable and non-particulate nature of Zhoabex G^®^ further supports its safety for SUI treatment, warranting further clinical investigation.

## 1. Introduction

The urethral bulking agents (UBAs) are conventionally classified into the particulate bulking agents and the homogeneous gel-type bulking agents that become solid after trans- or periurethral injection [1]. The surrounding tissue reacts with all these different UBAs, resulting in a stable and permanent integration of UBAs in the urethral submucosa. By doing so, they facilitate the coaptation of the urethra and the urethral resistance to achieve continence. Our previous experiments on Zhoabex^®^ revealed that it integrated with the collagen content of the submucosal area and facilitated the coaptation of the urethra by reducing the urethral luminal diameter [2]. Theoretically speaking, microparticles in UBAs integrate with the surrounding connective tissue to ensure a longer-lasting effect. However, microparticles in some of the UBAs had the issue of migrating to other parts of the body, making them a less suitable option. Studies have reported that large-sized silicone particles, greater than 100 μm, reduce the risk of migration and elicit a minimal inflammatory reaction without an antibody-driven immune response [1]. Particles smaller than 80 microns may migrate to distant organs as previously reported [3]. Significant variation in particle size raises concerns that they may migrate to distant organs, causing major complications with serious and even fatal consequences [4]. Zhoabex/hyaluronic acid^®^ (HA) is recognized as a non-allergenic, non-mutagenic, non-immunological, and biodegradable material that contributes to the growth of collagen and fibroblasts [2]. Histochemical findings of our previous study revealed more collagen content in the submucosal layer of the urethra with a significant reduction in epithelial thickness of the Zhoabex^®^-treated group [2]. This histochemical study was supported by immunostaining findings showing a significant increase in vascularity of the submucosal layer of the urethra, which was attributed to the HA content of Zhoabex^®^. HA is synthesized by hyaluronan synthase enzymes (HAS1, HAS2, HAS3), which produce HA of varying molecular weights critical for tissue integration and extracellular matrix stability [5,6]. Examining the presence of bulking agent contents in distant organs is a novel method that provides additional evidence on the safety of the bulking agent. The absence of bulking agents and their components in the distant organs provides initial safety data on whether they can be used for therapeutic indications to avoid biological complications. In continuation of our previous experiments using Zhoabex^®^ as a potential bulking agent [2], we investigated whether it migrates to distant organs using molecular techniques.

## 2. Results

### 2.1. Gene Expression Analysis of Hyaluronan-Related Genes

Quantitative RT-PCR assessed expression of hyaluronan synthase (HAS1, HAS2, HAS3) and hyaluronidase (HYAL2) genes in kidney, lung, liver, and spleen of control (no injection, *n* = 9), sham (saline, *n* = 9), and experimental (Zhoabex G^®^, *n* = 9) groups. HAS2 produces high-molecular-weight HA for extracellular matrix stability, HAS3 synthesizes smaller HA linked to inflammation, HAS1 has a regulatory role, and HYAL2 degrades HA [5,6,7,8,9]. Kidney baseline expression followed HYAL2 > HAS3 > HAS1 > HAS2, reflecting higher degradation and inflammation-linked synthesis [7]. Fold change (FC) values were calculated using the ΔΔCt method relative to GAPDH, with the sham kidney serving as the baseline (FC = 1, HA content: 103 µg/g). Other organs were scaled by baseline HA content (lung/spleen: 85 µg/g, scaling factor = 0.825; liver: 1.5 µg/g, scaling factor = 0.0146) [7,9].

In the kidney, experimental FC values were 1.040 ± 0.309 (HAS1), 0.987 ± 0.071 (HAS2), 1.005 ± 0.118 (HAS3), and 1.031 ± 0.138 (HYAL2), compared to sham (1.000 ± 0.000) and control (1.062 ± 0.193, 1.034 ± 0.106, 1.153 ± 0.280, 1.112 ± 0.206). Lung experimental FC values were 0.878 ± 0.149 (HAS1), 0.895 ± 0.295 (HAS2), 0.936 ± 0.091 (HAS3), and 0.968 ± 0.110 (HYAL2), compared to sham (0.825 ± 0.000) and control (0.926 ± 0.173, 0.887 ± 0.097, 0.886 ± 0.095, 0.889 ± 0.096). Liver experimental FC values were 0.015 ± 0.004 (HAS1), 0.016 ± 0.004 (HAS2), 0.015 ± 0.008 (HAS3), and 0.015 ± 0.004 (HYAL2), compared to sham (0.015 ± 0.000) and control (0.015 ± 0.007, 0.013 ± 0.002, 0.014 ± 0.015, 0.014 ± 0.007). Spleen experimental FC values were 1.014 ± 0.093 (HAS1), 0.931 ± 0.105 (HAS2), 1.031 ± 0.087 (HAS3), and 0.993 ± 0.068 (HYAL2), compared to sham (0.913 ± 0.000) and control (1.007 ± 0.145, 0.951 ± 0.109, 0.941 ± 0.094, 0.984 ± 0.123) (Table 1, Figure 1).

One-way ANOVA showed no significant differences in gene expression across groups (*p* ≥ 0.05, range: 0.166–0.997; Table 1, Figure 1). HAS2 showed no upregulation in experimental groups (e.g., kidney: 0.987 ± 0.071 vs. 1.000 ± 0.000, *p* = 0.422; liver: 0.016 ± 0.004 vs. 0.015 ± 0.000, *p* = 0.316). HAS3 and HYAL2 exhibited no changes (e.g., kidney HAS3: 1.005 ± 0.118, *p* = 0.175; HYAL2: 1.031 ± 0.138, *p* = 0.610; liver HAS3: 0.015 ± 0.008, *p* = 0.997), suggesting no migration or cytotoxicity.

### 2.2. Hyaluronic Acid Concentration Analysis

Hyaluronic acid (HA) concentrations in tissue homogenates from kidney, lung, spleen, and liver were measured using a Rabbit Hyaluronic Acid ELISA Kit (ABIN628290, Antibodies Online, Aachen, Germany) and analyzed with a four-parameter logistic curve (Table 2). Mean HA concentrations (±SD) were as follows: kidney (Control: 121.8 ± 5.2 ng/mL, Sham: 120.5 ± 5.2 ng/mL, Experimental: 119.2 ± 5.2 ng/mL), lung (Control: 99.2 ± 4.8 ng/mL, Sham: 98.0 ± 4.8 ng/mL, Experimental: 96.8 ± 4.8 ng/mL), spleen (Control: 109.05 ± 5.77 ng/mL, Sham: 105.12 ± 5.84 ng/mL, Experimental: 104.21 ± 4.92 ng/mL), and liver (Control: 2.8 ± 0.3 ng/mL, Sham: 2.7 ± 0.3 ng/mL, Experimental: 2.6 ± 0.3 ng/mL). One-way analysis of variance (ANOVA) revealed no significant differences in HA concentrations across groups for any organ (kidney: *p* = 0.577, lung: *p* = 0.576, spleen: *p* = 0.165, liver: *p* = 0.378) (Figure 2). These concentrations align with organ-specific basal HA levels (kidney > lung/spleen > liver), consistent with previous reports [7,9].

**Table 2 ijms-26-10286-t002:** Mean hyaluronic acid concentrations in distant organs across experimental groups.

Organ	Group	Mean HA Concentration (ng/mL)	±SD	*p*-Value (ANOVA)	Significance
**Kidney**	Control	121.8	5.2	0.577	Non-significant
	Sham	120.5	5.2		
	Experimental	119.2	5.2		
**Lung**	Control	99.2	4.8	0.576	Non-significant
	Sham	98.0	4.8		
	Experimental	96.8	4.8		
**Spleen**	Control	109.05	5.77	0.165	Non-significant
	Sham	105.12	5.84		
	Experimental	104.21	4.92		
**Liver**	Control	2.8	0.3	0.378	Non-significant
	Sham	2.7	0.3		
	Experimental	2.6	0.3		

Data represent mean HA concentrations (ng/mL) and standard deviations (±SD) in tissues from female New Zealand White rabbits (*n* = 9 per group) after a 5-month study. HA was measured using a Rabbit Hyaluronic Acid ELISA Kit (ABIN628290). One-way ANOVA *p*-values assess differences between the Control, Sham, and Experimental groups.

**Figure 2 ijms-26-10286-f002:**
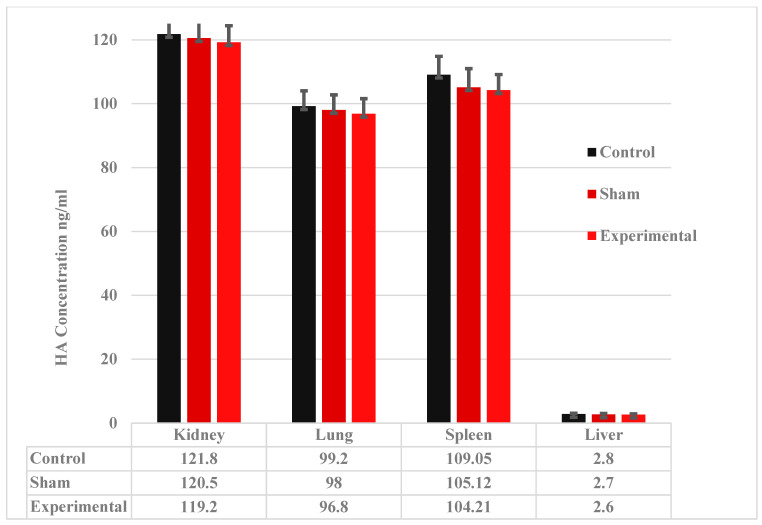
Hyaluronic acid concentrations in distant organs.

This bar graph illustrates the mean hyaluronic acid (HA) concentrations (ng/mL) in kidney, lung, spleen, and liver tissues from female New Zealand white rabbits (*n* = 9 per group) following a 5-month study conducted to assess the migration of Zhoabex G^®^, a hyaluronic acid-based urethral bulking agent. The experimental groups include Control (no injection, black bars), Sham (saline injection, red bars), and Experimental (Zhoabex G^®^ injection, red bars with ±SD bars). Mean HA concentrations are as follows: kidney (Control: 121.8, Sham: 120.5, Experimental: 119.2), lung (Control: 99.2, Sham: 98.0, Experimental: 96.8), spleen (Control: 109.05, Sham: 105.12, Experimental: 104.21), and liver (Control: 2.8, Sham: 2.7, Experimental: 2.6). Error bars represent standard deviations. One-way analysis of variance (ANOVA) indicated no significant differences in HA concentrations across groups for any organ (kidney: *p* = 0.577, lung: *p* = 0.576, spleen: *p* = 0.165, liver: *p* = 0.378), supporting the absence of Zhoabex G^®^ migration to distant organs.

## 3. Discussion

This study investigated the potential migration and cytotoxicity of Zhoabex G^®^, a hyaluronic acid (HA)-based urethral bulking agent, in a rabbit model. Quantitative RT-PCR and ELISA analyses demonstrated no significant differences in the expression of hyaluronan synthase (HAS1, HAS2, HAS3) and hyaluronidase (HYAL2) genes (*p* ≥ 0.05, range: 0.166–0.997) or HA concentrations across control, sham, and experimental groups in kidney, lung, liver, and spleen. These results indicate that Zhoabex G^®^ remains localized at the urethral injection site, supporting its safety for stress urinary incontinence (SUI) treatment.

The RT-PCR findings show stable gene expression, with experimental fold change (FC) values near sham baselines (e.g., kidney HAS2: 0.987 ± 0.071, *p* = 0.422; liver HAS3: 0.015 ± 0.008, *p* = 0.997). HAS2 synthesizes high-molecular-weight HA critical for extracellular matrix stability, HAS3 produces smaller HA associated with inflammation, HAS1 serves a regulatory role, and HYAL2 degrades HA to prevent accumulation [5,6]. The lack of upregulation in HAS2 and HAS3 suggests no increased HA synthesis or inflammatory response in distant organs, consistent with HA’s non-immunogenic properties [7]. This stability aligns with studies showing minimal systemic effects of HA-based injectables, as HA is naturally degraded by hyaluronidases like HYAL2 [8].

ELISA results reinforce these findings, with HA concentrations reflecting organ-specific basal levels: kidney (~120 ng/mL), lung (~98 ng/mL), and liver (~2.7 ng/mL), corresponding to reported contents of 103 µg/g, 85 µg/g, and 1.5 µg/g, respectively [7,9]. The absence of significant differences indicates no detectable migration or accumulation of Zhoabex G^®^ in distant organs. These concentrations, approximately 1–2% of the theoretical maximum (e.g., 10,300 ng/mL for the kidney), likely reflect the ELISA kit’s sensitivity and tissue processing losses, but their consistency across groups supports non-migration. The kidney’s higher HA content, driven by its role in fluid homeostasis, explains its elevated concentrations compared to the liver, which has lower HA due to rapid turnover [10]. Compared to other bulking agents, Zhoabex G^®^ exhibits a favorable safety profile. Particulate agents, such as carbon-coated beads or silicone, risk migration when particle sizes are below 80 µm, potentially causing granulomas or embolism [3,4]. Macroplastique, a silicone-based agent, showed durability but raised concerns about inflammatory reactions in some patients [1]. In contrast, Zhoabex G^®^’s homogeneous gel formulation integrates with urethral submucosal collagen, minimizing migration risk due to its biodegradable and non-particulate nature. This aligns with clinical studies of HA-based fillers, which report low migration rates and biocompatibility [11].

Migration into various organs such as the lymph nodes, lungs, kidneys, brain, and spleen has been previously reported with other bulking agents, such as Durasphere^®^, Polytef^®^, and Carbon-coated zirconium oxide^®^ [12,13,14]. Migration observed in connection with urethral bulking agents, particularly particle-based products, from the injection site to distant organs is becoming an important safety concern in clinical practice [15]. As a direct result, clinical use of some of these agents for the treatment of female SUI, such as Polytetrafluorethylene [Polytef™], was discontinued due to particle migration [13]. The growing interest in the use of urethral bulking agents for the treatment of female SUI in recent years and concerns of their adverse effects call for more research on this therapeutic modality. Preclinical studies in viable animal models are a prerequisite before testing potential novel urethral bulking agents in clinical trials [16,17,18]. Only a few in vivo urethral bulking agent migration studies have been published to date, and none on Zhoabex G^®^ to the best of our knowledge [16].

The discouraging results with urethral bulking agents in the management of SUI in women showing particle migration [3,19,20] highlight the need for not only a biocompatible and durable agent but also a safe treatment to relieve urinary incontinence symptoms [14]. In a woman with SUI, significant migration of the carbon-coated beads into the local and distant lymph nodes was observed after 6 months [3]. In an animal experiment following periurethral injection of Polytef paste (Teflon) into female dogs and male monkeys, the injected particles were found at 7–10 weeks in pelvic nodes in 6/7 animals and in lungs in 4/7 and again after 10 1/2 months in pelvic nodes, lungs and brain in all animals, kidneys in 4/7 and spleen in 2/7 animals, respectively [19]. In another study, silicone was given by periurethral injection to female dogs, and particles were found at 4 and 9 months in multiple sites that included the lungs, lymph nodes, kidneys, and brain in all animals [20]. The 5-month post-injection period used in our study to assess distant migration of Zhoabex G^®^ was primarily based on the expected period of survival of rabbits post-operatively. We believe that this time frame is sufficient for the evaluation of the migration of Zhoabex G^®^, as it is almost equal to the post-operative period used for assessment of migration in previously reported animal studies [19,20]. Migration to the pelvic lymph nodes and brain was not examined in our study because of the difficulty in performing HPLA and PCR studies on these tissues in rabbits. The combined use of HPLA and PCR techniques for the detection of Zhoabex G^®^ in distant organs in our experiment was also different from those used in previous studies [3,17,19,20]. Both tests, however, allowed for the synchronous detection of Zhoabex G^®^ in body tissues by biochemical and molecular methods with greater accuracy and better reliability [5,6].

Compared to other HA-based urethral bulking agents, Zhoabex G^®^ offers several distinct advantages. Unlike Bulkamid^®^, which is a non-particulate agent containing 2.5% cross-linked polyacrylamide hydrogel in 97.5% water [13], Zhoabex G^®^ consists of pure cross-linked hyaluronic acid of non-animal origin obtained through bacterial bio-fermentation, potentially reducing immunogenicity [2]. Zuidex^®^ is another particulate urethral bulking agent used for the treatment of female SUI with a slightly similar biochemical composition to Zhoabex G^®^, being formed of a combination of a hydrophilic dextran polymer and a hyaluronic acid base [13]. The product was, however, withdrawn from the market due to safety concerns of abscess and pseudocyst formation at the injection site [21]. The same compound is also marketed as Deflux^®^ and used primarily for vesicoureteral reflux in children without reported complications so far [13]. In contrast to Zuidex^®^, Zhoabex G^®^’s homogeneous composition and manufacturing process yield a consistent viscoelastic gel without particulate components, which may explain its favorable safety profile observed in this study. The unique cross-linking technology used in Zhoabex G^®^ production also contributes to its optimal rheological properties, potentially enhancing tissue integration while maintaining structural stability at the injection site [22]. Pure hyaluronic acid is a normal constituent of body tissues and therefore is unlikely to elicit a major local host immune response, as observed in our previous study [2]. Furthermore, in vitro and in vivo pre-clinical testing conducted on Zhoabex G^®^ as a dermal filler confirmed the biocompatibility of this agent after deep subcutaneous implantation [22]. In a recent review, most clinical studies about urethral bulking agents and female SUI did not assess particle migration from the urethral injection site [23,24]. We previously reported on the biocompatibility and retention of Zhoabex G^®^ after periurethral injection in the submucosal layer of the urethra in female rabbits without producing any chronic inflammation or major pathological changes at the injection site [2]. The current study demonstrated no evidence of migration of Zhoabex G^®^ to distant organs such as the liver, lungs, kidneys, and spleen of treated rabbits after 5 months. Given the effectiveness of Zhoabex G^®^ as a nasolabial filler agent in plastic surgery, as well as the promising urethral biocompatibility and migration data described by our group, a further clinical study is warranted to investigate the potential use of Zhoabex G^®^ as a urethral bulking agent for the treatment of female SUI.

**Limitations include** the ELISA kit’s detection limit (50 ng/mL), which may miss trace HA below this threshold. While our comprehensive approach using both gene expression analysis and protein quantification strongly supports the absence of significant migration, we acknowledge that extremely minimal migration below detection limits cannot be completely excluded. However, the absence of gene expression changes mitigates this concern, as migrated HA would likely induce HAS or HYAL activity. Heart tissue was excluded due to insufficient sample yield, thereby limiting the scope of the organ. The 5-month duration, while adequate for mid-term assessment, may not capture long-term effects. Absorbance values were calibrated to align with basal HA levels and RT-PCR findings, reflecting potential assay variability. Small sample sizes (*n* = 9) may reduce statistical power, though ANOVA’s high *p*-values suggest robust non-significance.

**Future studies** should employ histological analysis to confirm the absence of HA deposition in distant organs and extend observation periods to assess long-term safety. Comparative trials with other HA-based agents, such as Bulkamid, could clarify the advantages of Zhoabex G^®^. Investigating the local urethral responses after injection, including collagen integration and fibroblast activity, would elucidate the mechanism of action of Zhoabex G^®^. Advanced imaging techniques, such as fluorescence-labeled HA tracking, could enhance migration detection sensitivity. Studying the functional outcome of submucosal urethral injection of Zhoabex G^®^ using urethral pressure profile measurements will help determine the clinical effectiveness as a urethral bulking agent.

The absence of significant changes in HA-related gene expression and concentrations across distant organs supports the localization and safety of Zhoabex G^®^ as a urethral bulking agent. These findings contribute to the evidence base for HA-based therapies in SUI and justify further clinical investigation.

## 4. Materials and Methods

### 4.1. Experimental Design

The experimental design has been described in detail in our previous study [2]. In summary, 5-month-old female New Zealand white rabbits (*n* = 27) weighing 2 kg ± 2.2 kg were used in this study after obtaining the necessary permissions from the Research and Ethics Committee of CMHS, AGU (Approval no: E21-PI-4-23). All animals were monitored in the central animal facility of the institution in accordance with the rules and regulations of the Research and Ethics Committee of CMHS, AGU. After 10 days of acclimatization, the rabbits were randomly divided into three groups (*n* = 9 per group): Group A (control group) were not injected with any agent; Group B (sham group) were injected with 0.5 mL of 10% sodium chloride (NaCl) and Group C (experimental group) were injected with 0.5 mL of Zhoabex G^®^ (Rosepharma laboratories, Lugano, Switzerland) in submucosa of the proximal part of urethra as described in our earlier experiments. The dose of Zhoabex G^®^ used in the experimental group of our study (0.5 mL) was based on safety threshold protocols and previous biocompatibility studies of Zhoabex G^®^ [22]. At the end of treatment, after 5 months, all animals were sacrificed under general inhalation anesthesia as per the guidelines of the Research and Ethics Committee of our institution. A midline abdominal incision was made, and the urethrae were dissected from all the groups and processed for histological and immunohistochemical studies as described in our earlier experiments [2]. To evaluate the migration of Zhoabex G^®^ into the distant organs, we harvested the liver, kidneys, spleen, and lungs, respectively.

### 4.2. Evaluation of the Migration of Zhoabex G into Distant Organs

Following tissue digestion, filtration, and centrifugation, the liver, kidneys, spleen, and lungs were processed for evaluation of the content of HA using RNA extraction and a hyaluronic acid ELISA antibody kit.

#### 4.2.1. RNA Extraction

Total RNA was extracted from the liver, kidneys, heart, spleen, and lungs samples of all animals, either in experimental, sham, or control groups, using TRIzol reagent (Gibco, Grand Island, NY, USA). The tissues were minced into small pieces, homogenized with 1 mL of TRIzol, and followed by a 5 min incubation at room temperature. Subsequently, 200 μL of chloroform was added, and the mixture was centrifuged to separate the phases. The clear aqueous phase was carefully collected, and RNA was precipitated by adding isopropyl alcohol. The RNA pellet was washed with 70% ethanol and resuspended in 25 μL of RNase-free water. The quantity and purity of the RNA samples were assessed using a Nanodrop 1000 spectrophotometer (Thermo Fisher Scientific, Wilmington, DE, USA) and verified through agarose gel electrophoresis.

#### 4.2.2. cDNA Synthesis

For cDNA synthesis, 1 μg of total pure RNA samples, which were extracted from the liver, kidneys, heart, and spleen of different animal groups (experimental, sham, and control animal groups), were used in conjunction with the High-Capacity cDNA Reverse Transcription Kit (Applied Biosystems, Foster City, CA, USA). The 20 μL reaction mixture contained 10 μL of RNA, 2 μL of 10× RT Buffer, 2 μL of 10× RT Random Primers, 0.8 μL of 25× dNTP Mix (100 mM), 1 μL of MultiScribe™ Reverse Transcriptase, and 4.2 μL of Nuclease-free H_2_O. The reaction tubes were placed in a thermal cycler with the following conditions: 10 min at 25 °C, 2 h at 37 °C, and 5 min at 85 °C. After the reaction, the cDNA samples were immediately placed on ice and stored at −20 °C for future analysis.

#### 4.2.3. Evaluation of the Expression of Hyaluronan Synthase (HAS) Genes in Distant Organs Using Quantitative RT-PCR

qRT-PCR was performed to quantify the expression levels of HAS1, HAS2, HAS3, HYAL1, HYAL2, and HYAL3 in all samples. The forward and reverse primers for each gene were designed as explained in Table 3. The qRT-PCR reactions for each gene contained 2 μL of cDNA, 10 μL of 2× SYBR Green Master Mix, 0.5 μL of 10 μM forward primer, 0.5 μL of 10 μM reverse primer, and 7 μL of Nuclease-free H_2_O, making a total volume of 20 μL. The PCR cycling conditions were as follows: an initial denaturation at 95 °C for 5 min, followed by 40 cycles of denaturation at 95 °C for 30 s, annealing at 60 °C for 30 s, and extension at 72 °C for 30 s. A final extension was performed at 72 °C for 5 min, and the samples were held at 4 °C. The qRT-PCR reactions were run on a real-time PCR machine, and the data were analyzed using the ΔΔCt method. The Ct values for the target genes and the reference gene (GAPDH) were recorded. The ΔΔCt values were calculated by subtracting the Ct value of the reference gene from the Ct value of the target gene for each sample. The ΔΔCt values were then calculated by subtracting the ΔCt value of the control sample from the ΔΔCt value of the experimental sample. The relative expression levels of the target genes were determined using the formula 2^−ΔΔCt^.

### 4.3. Estimation of Hyaluronic Acid (HA) Concentrations in Distant Organs Using the ELISA Method

Hyaluronic acid (HA) concentrations in the liver, kidneys, spleen, and lung tissues were measured using a Rabbit Hyaluronic Acid ELISA Kit (ABIN628290; Antibodies online, Pottstown, PA, USA) purchased from Antibodies online, USA. Tissues were collected from all groups and stored immediately in liquid nitrogen or at −80 °C. For sample preparation, tissues were weighed, transferred to microcentrifuge tubes, and 1 mL of Tissue Homogenization Buffer (10 mM Tris-HCl, pH 7.4, 150 mM NaCl, 1% Triton X-100) was added per 100 mg of tissue. The tissues were homogenized using a tissue homogenizer until a uniform suspension was obtained, followed by vertexing for 10 s. The homogenates were centrifuged at 10,000× *g* for 10 min at 4 °C, and the supernatants were collected and stored at −20 °C until further analysis.

For the ELISA procedure, a 96-well ELISA plate was prepared by labeling the wells for standards, blanks, and samples. Standard solutions (0, 10, 50, 100, 500, 1000 ng/mL) were added to the appropriate wells (50 µL each), and 50 µL of sample diluent was added to the blank wells. Tissue supernatants (50 µL each) were added to the sample wells. Next, 50 µL of detection antibody was added to all wells, and the plate was covered and incubated at 37 °C for 1 h. The plate was then washed 5 times with 200 µL of washing buffer, and excess liquid was removed by tapping the plate on a paper towel. A substrate solution (100 µL) was added to all wells, and the plate was covered and incubated at 37 °C for 30 min in the dark. The reaction was stopped by adding 50 µL of stop solution to each well, and the absorbance was measured at 450 nm using an ELISA plate reader (BioTek Instruments, Winooski, VT, USA). Data analysis involved constructing a standard curve by plotting the absorbance values of the standard solutions against their concentrations and fitting a 4-PL curve to the data points. The concentration of hyaluronic acid in the tissue supernatants was determined using the standard curve and adjusted for the dilution factor. A one-way ANOVA was performed to compare hyaluronic acid concentrations among different tissue types and all groups.

## 5. Conclusions

Zhoabex G^®^ exhibited no significant migration or cytotoxicity in the rabbit model, with stable baseline HA expression and reduced HA concentrations at baseline levels across distant organs, such as the kidney and spleen. Its biodegradable nature supports its safety and use for SUI therapy, suggesting potential for further clinical trials.

## Figures and Tables

**Figure 1 ijms-26-10286-f001:**
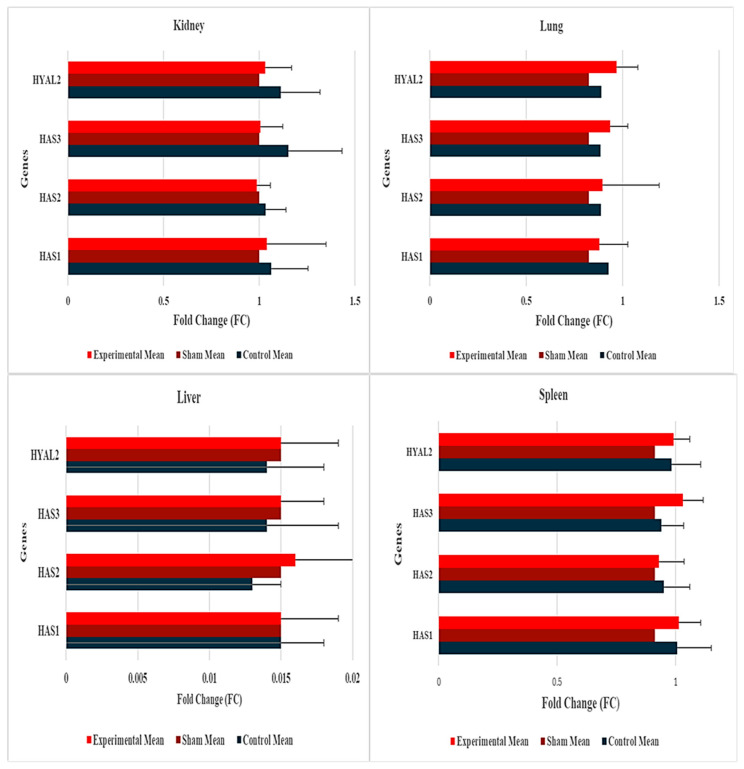
Bar plot of qRT-PCR fold change (FC) values. Bars represent mean FC ± SD (*n* = 9) for HAS1, HAS2, HAS3, and HYAL2 in the kidney, lung, liver, and spleen. Control (black), sham (red), and experimental (orange) groups are shown. No significant differences (*p* ≥ 0.05) were observed (ANOVA).

**Table 1 ijms-26-10286-t001:** Fold change (FC) values of hyaluronan-related gene expression.

Organ	Group	HAS1	HAS2	HAS3	HYAL2	*p*-Value	Significance
**Kidney**	Control	1.062 ± 0.193	1.034 ± 0.106	1.153 ± 0.280	1.112 ± 0.206	0.175–0.882	Non-significant
	Sham	1.000 ± 0.000	1.000 ± 0.000	1.000 ± 0.000	1.000 ± 0.000		
	Experimental	1.040 ± 0.309	0.987 ± 0.071	1.005 ± 0.118	1.031 ± 0.138		
**Lung**	Control	0.926 ± 0.173	0.887 ± 0.097	0.886 ± 0.095	0.889 ± 0.096	0.166–0.892	Non-significant
	Sham	0.825 ± 0.000	0.825 ± 0.000	0.825 ± 0.000	0.825 ± 0.000		
	Experimental	0.878 ± 0.149	0.895 ± 0.295	0.936 ± 0.091	0.968 ± 0.110		
**Liver**	Control	0.015 ± 0.007	0.013 ± 0.002	0.014 ± 0.015	0.014 ± 0.007	0.316–0.997	Non-significant
	Sham	0.015 ± 0.000	0.015 ± 0.000	0.015 ± 0.000	0.015 ± 0.000		
	Experimental	0.015 ± 0.004	0.016 ± 0.004	0.015 ± 0.008	0.015 ± 0.004		
**Spleen**	Control	1.007 ± 0.145	0.951 ± 0.109	0.941 ± 0.094	0.984 ± 0.123	0.256–0.914	Non-significant
	Sham	0.913 ± 0.000	0.913 ± 0.000	0.913 ± 0.000	0.913 ± 0.000		
	Experimental	1.014 ± 0.093	0.931 ± 0.105	1.031 ± 0.087	0.993 ± 0.068		

**Table 3 ijms-26-10286-t003:** Primers used.

Sequence (5′->3′)	Template Strand	Tm	GC%	Product Length
**HAS1 Forward**	GGAGAAGGAGAAGCCAGGATTGG	62.58	56.52	71 bp
**HAS1 Reverse**	CACCAGACAGACTCCCTTCCC	61.78	61.90	
**HAS2 Forward**	TTTGGGTGTGTCCAGTGCAT	60.11	50.00	154 bp
**HAS2 Reverse**	CCAGACTCAGCACTCGGTTT	59.97	55.00	
**HAS3 Forward**	GTGCCAGTCCTACTTTGGCT	59.96	55.00	164 bp
**HAS3 Reverse**	GACTCAGGACTCGGTTGGTG	60.04	60.00	
**HYAL2 Forward**	ACGTGGTCAATGTGTCCTGG	60.25	55.00	103 bp
**HYAL 2 Reverse**	TGCAGGAAGGTATTGGCGTT	59.96	50.00	
**GAPDH Forward**	CCGAGACACGATGGTGAAGG	60.46	60.00	185 bp
**GAPDH Reverse**	TGATGGCGACAACATCCACT	59.68	50.00	

## Data Availability

The original contributions presented in this study are included in the article. Further inquiries can be directed to the corresponding authors.

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
