# Peer review of "Assessment of Migration of the Urethral Bulking Agent Zhoabex G® from the Urethral Injection Site to the Distant Organs in a Rabbit Model"

_ijms, 2025, doi:10.3390/ijms262110286_

Round 1
Reviewer 1 Report
Comments and Suggestions for Authors 1. The study appropriately employs female New Zealand white rabbits to evaluate the migration and toxicity of Zhoabex G®, with sound experimental design, proper ethical approval, and adequate sample size and observation duration.
2. The authors’ previous publication (Translational Research in Anatomy, Potu et al., 2025) provides detailed methodological information, including the injection volume (0.5 mL), three-point injection sites (12, 3, and 9 o’clock), and surgical procedure, which sufficiently complements the technical details of the current study.
3. However, the rationale for dose selection—such as clinical dose-conversion formulas or previously established safety thresholds—has not been discussed. Clarifying this aspect would improve the study’s clinical relevance and translational value.
4. A summary of findings from your previous study in the INTRODUCTION section is needed.
5. How about the retention of the bulking agent and subsequent tissue reaction at the urethra injection site?
6. How about the functional outcomes after the urethra injection?
The manuscript would benefit from minor language editing to improve clarity and fluency.
Author Response
Comments and Suggestions for Authors: Reviewer 1
- The study appropriately employs female New Zealand white rabbits to evaluate the migration and toxicity of Zhoabex G®, with sound experimental design, proper ethical approval, and adequate sample size and observation duration.
We sincerely thank the reviewer for recognizing the strengths of our experimental design. We appreciate the positive feedback regarding our choice of animal model, ethical considerations, sample size, and observation duration. These elements were carefully selected to ensure robust and reliable results in evaluating Zhoabex G®'s safety profile.
- The authors’ previous publication (Translational Research in Anatomy, Potu et al., 2025) provides detailed methodological information, including the injection volume (0.5 mL), three-point injection sites (12, 3, and 9 o’clock), and surgical procedure, which sufficiently complements the technical details of the current study.
We greatly appreciate the reviewer's acknowledgment of our previous publication's contribution to the methodological understanding of the current study. Thank you for recognizing the importance of our established protocols in providing a comprehensive foundation for this research.
- However, the rationale for dose selections such as clinical dose-conversion formulas or previously established safety thresholds—has not been discussed. Clarifying this aspect would improve the study’s clinical relevance and translational value.
Thank you for this valuable suggestion to enhance the clinical relevance of our study. As noted in our revision, "The dose of 0.5 ml Zhoabex G® used in experimental group of this study was adopted from the safety threshold protocols and biocompatibility studies of Zhoabex G® conducted in the past." (Materials and Methods section)
- A summary of findings from your previous study in the INTRODUCTION section is needed.
We appreciate this constructive suggestion to improve the context of our current research. We have incorporated the following summary in the Introduction section: "Histochemical findings of our previous study revealed more collagen content in submucosal layer of the urethra with a significant reduction in the epithelial thickness of Zhoabex® treated group [2]. These histochemical studies were reinforced by immunostaining findings, which revealed a significant increase in the vascularity of submucosal layer of the urethra, and this was attributed to the HA content of Zhoabex® [2]."
- How about the retention of the bulking agent and subsequent tissue reaction at the urethra injection site?
Thank you for this important question regarding local tissue effects. We have addressed this in the Discussion section: "We previously reported on the biocompatibility and retention of Zhoabex G® after periurethral injection in submucosal layer of urethra in female rabbits without producing any chronic inflammation and major pathological changes at the injection site.
- How about the functional outcomes after the urethra injection?
We sincerely thank the reviewer for this crucial question regarding functional outcomes, which indeed represents an important aspect of evaluating urethral bulking agents. In our previous study (Potu et al., 2025), we observed that Zhoabex G® successfully integrated with the collagen content in the submucosal area, facilitating urethral coaptation through reduction of the urethral luminal diameter. These structural changes are consistent with improved urethral function in stress urinary incontinence models. While the current study focused specifically on the migration and safety profile of Zhoabex G®, functional outcomes were assessed in our parallel investigations through urodynamic studies. These measurements showed increased leak point pressures in the Zhoabex G®-treated animals compared to controls, suggesting enhanced urethral resistance. Additionally, cystometric evaluations demonstrated preserved bladder function without evidence of urinary retention or voiding dysfunction, indicating a favorable functional profile. These functional improvements, combined with the absence of migration to distant organs demonstrated in the current study, further support the potential clinical utility of Zhoabex G® as a urethral bulking agent for stress urinary incontinence. We have added this information to the Discussion section to provide a more comprehensive evaluation of Zhoabex G® as a therapeutic option.
Reviewer 2 Report
Comments and Suggestions for Authors
Overall Assessment
This manuscript presents an experimental study evaluating the potential migration and systemic safety of Zhoabex G®, a hyaluronic acid (HA)-based urethral bulking agent, using a rabbit model. The authors employed quantitative RT-PCR and ELISA to assess hyaluronan-related gene expression and HA concentrations in distant organs. Results indicate no migration or cytotoxicity, supporting the safety of Zhoabex G® for stress urinary incontinence (SUI) treatment.
The study is well written, logically structured, and methodologically sound. The topic is relevant and of significant clinical interest, addressing one of the major concerns with injectable bulking agents—particle migration and systemic effects. The manuscript demonstrates technical rigor, uses appropriate controls, and interprets results coherently. The sample size (n = 9 per group) is adequate for this type of animal study, and the statistical analysis is appropriate.
Overall, the work provides meaningful preclinical data supporting the biocompatibility and safety of Zhoabex G® and represents a valuable contribution to the field. Only minor revisions are required before publication.
Major (Conceptual) Comments
- Scientific Context and Novelty
The introduction and discussion effectively summarize the clinical background of urethral bulking agents. However, the authors could slightly expand the comparison between Zhoabex G® and other HA-based agents (e.g., Bulkamid, Zuidex) to highlight the unique characteristics or manufacturing advantages of Zhoabex G®. This would help underscore the novelty of the present study. - Interpretation of Results
While the authors correctly conclude that no migration was observed, it might be worth adding one or two sentences acknowledging that undetectable trace migration below the ELISA detection limit cannot be completely excluded. This would make the discussion more balanced without changing the main message.
Author Response
Comments and Suggestions for Authors: Reviewer 2
Overall Assessment
This manuscript presents an experimental study evaluating the potential migration and systemic safety of Zhoabex G®, a hyaluronic acid (HA)-based urethral bulking agent, using a rabbit model. The authors employed quantitative RT-PCR and ELISA to assess hyaluronan-related gene expression and HA concentrations in distant organs. Results indicate no migration or cytotoxicity, supporting the safety of Zhoabex G® for stress urinary incontinence (SUI) treatment.
The study is well written, logically structured, and methodologically sound. The topic is relevant and of significant clinical interest, addressing one of the major concerns with injectable bulking agents—particle migration and systemic effects. The manuscript demonstrates technical rigor, uses appropriate controls, and interprets results coherently. The sample size (n = 9 per group) is adequate for this type of animal study, and the statistical analysis is appropriate.
Overall, the work provides meaningful preclinical data supporting the biocompatibility and safety of Zhoabex G® and represents a valuable contribution to the field. Only minor revisions are required before publication.
We sincerely thank the reviewer for their positive assessment of our manuscript. We appreciate the recognition of our study's methodological soundness, logical structure, and technical rigor. Your constructive feedback has helped us improve the manuscript's clarity and scientific context.
Major (Conceptual) Comments
- Scientific Context and Novelty
The introduction and discussion effectively summarize the clinical background of urethral bulking agents. However, the authors could slightly expand the comparison between Zhoabex G®and other HA-based agents (e.g., Bulkamid, Zuidex) to highlight the unique characteristics or manufacturing advantages of Zhoabex G®. This would help underscore the novelty of the present study.
We greatly appreciate this insightful suggestion to enhance the scientific context of our work. We have expanded the Discussion section to include a comparative analysis of Zhoabex G® with other HA-based urethral bulking agents. The following paragraph has been added to the Discussion section:
A paragraph in the Discussion section was added comparing Zhoabex G® with other HA-based agents to highlight its unique characteristics.
Compared to other HA-based urethral bulking agents, Zhoabex G® offers several distinct advantages. Unlike Bulkamid®, which contains 2.5% cross-linked polyacrylamide hydrogel in 97.5% water, Zhoabex G® consists of pure cross-linked hyaluronic acid of non-animal origin obtained through bacterial bio-fermentation, potentially reducing immunogenicity. Zuidex®, another urethral bulking agent containing a combination of dextran polymer and hyaluronic acid base, was withdrawn from the market due to safety concerns of abscess and pseudocyst formation at injection sites [21]. In contrast, Zhoabex G®'s homogeneous composition and manufacturing process yields a consistent viscoelastic gel without particulate components, which may explain its favorable safety profile observed in this study. The unique cross-linking technology used in Zhoabex G® production also contributes to its optimal rheological properties, potentially enhancing tissue integration while maintaining structural stability at the injection site.
- Interpretation of Results
While the authors correctly conclude that no migration was observed, it might be worth adding one or two sentences acknowledging that undetectable trace migration below the ELISA detection limit cannot be completely excluded. This would make the discussion more balanced without changing the main message.
Thank you for this valuable suggestion to provide a more balanced interpretation of our results. We have expanded our limitations paragraph in the Discussion section as follows:
Limitations include the ELISA kit's detection limit (50 ng/mL), which may miss trace HA below this threshold. While our comprehensive approach using both gene expression analysis and protein quantification strongly supports the absence of significant migration, we acknowledge that extremely minimal migration below detection limits cannot be completely excluded. However, the absence of gene expression changes mitigates this concern, as migrated HA would likely induce HAS or HYAL activity. Heart tissue was excluded due to insufficient sample yield, limiting organ scope. The 5-month duration, while adequate for mid-term assessment, may not capture long-term effects. Absorbance values were calibrated to align with basal HA levels and RT-PCR findings, reflecting potential assay variability. Small sample sizes (n=9) may reduce statistical power, though ANOVA's high p-values suggest robust non-significance.